# Large-Scale Portfolio Optimization Using Biogeography-Based Optimization

**Wendy Wijaya *** and **Kuntjoro Adji Sidarto**

Department of Mathematics, Faculty of Mathematics and Natural Sciences, Institut Teknologi Bandung, Ganesa Street No. 10, Bandung 40132, Indonesia; sidarto@itb.ac.id
* Correspondence: w3nd1.itc@gmail.com

**Abstract:** Portfolio optimization is a mathematical formulation whose objective is to maximize returns while minimizing risks. A great deal of improvement in portfolio optimization models has been made, including the addition of practical constraints. As the number of shares traded grows, the problem becomes dimensionally very large. In this paper, we propose the usage of modified biogeography-based optimization to solve the large-scale constrained portfolio optimization. The results indicate the effectiveness of the method used.

**Keywords:** biogeography-based optimization; constrained optimization; mean-variance model

## 1. Introduction

A portfolio can be defined as a collection of assets, which can include cash, real estate, stocks, or crypto. Portfolio optimization concerns maximizing returns and minimizing risks; returns are the expected profit from the investment, whereas risks are the possible changes in values of the investment. In Markowitz (1952), the authors proposed the use of the means and variances of the portfolio as the return and risk measures. Good practice in portfolio optimization is very crucial in investment, because it greatly affects the outcome of the investment. In this paper, we focus on portfolios consisting of correlated stocks.

The model proposed by Markowitz (1952) has been studied by many researchers over the years. A possible approach for the problem is the capital asset pricing model (CAPM). CAPM has been used extensively by many financial practitioners. In Parmikanti et al. (2020), the authors studied portfolio optimization under CAPM with a heteroscedastic model for the return series. Since its introduction, many improvements have been made to the model. Some improvements add practical constraints, which can be discrete or continuous; hence, the resulting model becomes mixed-integer nonlinear programming (MINLP). For instance, Jobst et al. (2001) and Bartholomew-Biggs and Kane (2009) considered adding roundlot constraints, which means that shares must be bought in a multiple of some integer. In Jobst et al. (2001), the authors used the FortMP solver, whereas Bartholomew-Biggs and Kane (2009) used DIRECT hybridized with quasi-Newton methodology. In Jobst et al. (2001), the authors also noted that the resulting efficiency is not continuous, making CAPM inapplicable in this case. With increasing complexity, various techniques also emerged. AUGMECON2 is a state-of-the-art multi-objective MINLP solver. It was introduced by Mavrotas and Florios (2013) and has been shown to very effective in solving multi-objective MINLP. Recent use of AUGMECON2 in portfolio optimization can be found in Chen et al. (2021). The downside of this method is its computational complexity. In Chen et al. (2021), the authors noted that for some large-dimensional problems, AUGMECON2 did not give a converged solution after 7 days.

To circumvent the complexity of exact methods, an efficient optimizer is needed. One popular approach is to use metaheuristic algorithms. Metaheuristic algorithms are usually inspired by natural processes in biology, chemistry, physics, or society. Most of the time, it is expected that metaheuristic algorithms can produce near-optimal solutions.

Furthermore, an exact method is expected to be implemented later to obtain more accuracy, such in Bartholomew-Biggs and Kane (2009). In Chang et al. (2000), the authors proposed a series of modified metaheuristic algorithms that exploit the structure of the MV model with cardinality and quantity constraints. The metaheuristic algorithms used were genetic algorithm (GA), tabu search (TS), and simulated annealing (SA). To handle the cardinality and quantity constraints, they implemented an algorithm to adjust the solutions.

Another example of a metaheuristic algorithm is biogeography-based optimization (BBO). It was developed by Simon (2008); its inspiration is the dynamics of the geography of habitats. It basically consists of migration and mutation. Elitism is also added to ensure faster convergence. BBO has been used to solve numerous optimization problems in the real world. Some newer applications of BBO can be found in Reihanian et al. (2023) and Ren et al. (2023). Those studies concluded that BBO is a very powerful optimizer. For MINLP, Garg (2015) showed the efficiency of BBO in solving reliability problems and the results demonstrated the superiority of BBO over other metaheuristics. In their approach, the integral and discrete constraints were treated as if they were continuous, but in the function evaluation, they were rounded accordingly. This is sensible, because BBO is known for its effectiveness in continuous optimization. One of the reasons why we chose BBO is that it requires minimal parameters and is easy to implement.

For applications in portfolio optimization, there are some entries in the literature that have used BBO as the main optimizer. In Ye et al. (2017), the authors used BBO to solve a portfolio optimization problem with second-order stochastic dominance constraints. In Garg and Deep (2019), the authors used a variant of BBO called Laplacian biogeogeography-based optimization (LX-BBO) to find portfolio allocation from 10 assets in an MV model. In Panwar et al. (2018), the authors used BBO to solve a constrained MV model and applied the results in forecasting via Monte Carlo. The number of assets used in that research was 15.

Over the time, the number of companies listed in the stock markets are increasing. There are many markets with a very large number of companies. Although this provides a good opportunity for investors to choose assets, this also creates the problem in choosing suitable assets. In Perold (2022), the author studied how to efficiently choose a subset of a large set to optimize a portfolio. He considered a constrained portfolio optimization with a cardinality constraint and a quantity constraint. His method was inspired by quadratic programming techniques and was later improved to work very well for solving portfolio optimization. In Qu et al. (2017), the authors considered a multiobjective constrained mean-variance model and used four methods to solve the problem. The methods they used were Normalized Multiobjective Evolutionary Algorithm based on Decomposition (NMOEA/D), Multiobjective Differential Evolution based on Summation Sorting (MODE-SS), and Multiobjective Differential Evolution based on Nondomination Sorting (MODE-NDS), Multiobjective Comprehensive Learning Particle Swarm Optimizer (MO-CLPSO), and Nondominated Sorting Genetic Algorithm II (NSGA-II). The constraint they added to the model was a preselection constraint. They concluded that the methods were efficient for large-scale portfolio optimization. They also suggested adding practical constraints such as the cardinality constraint and the quantity constraint for further research.

In this paper, we proposed the usage of the heuristic ideas of Chang et al. (2000) but implemented in a BBO framework to solve a constrained MV model. The reason we used the ideas of Chang et al. (2000) is that they worked very well on a large-scale portfolio in their study. The dimensions studied by that work were 31, 85, 89, 98, and 225. Their methods could solve a large-scale portfolio optimization problem with high accuracy and in short time. It is clear that standard methods do not solve this problem effectively because the computational complexity is very large.

We used data from ORLibrary, which is available online. The same data were used in Chang et al. (2000) and Kabbani (2022). We also compare our results with theirs using the same performance metric. The results show that BBO is competitive in comparison to other methods.

The organization of this paper is as follows. Section 2 introduces the problems that we solve in this paper and how we solve them. The problem is multiobjective constrained portfolio optimization. Then, introduce biogeography-based optimization (BBO) before detailing the method we propose. Section 3 contains the results of our proposed approach and a comparison with other studies. Conclusions and further improvements are also included in that section.

## 2. Materials and Methods

### 2.1. Portfolio Optimization

The aim of this subsection is to introduce the problems discussed in the paper. First, we describe the unconstrained MV model. Then, we discuss the constrained MV model. We follow the formulation used in Kabbani (2022).

#### 2.1.1. Unconstrained MV Model

Suppose there are $n$ assets, $A_1, A_2, \ldots, A_n$. Let $x_1, x_2, \ldots, x_n$ denote the budget share or allocation for the assets. The unconstrained MV model can be written as

$$\min \sum_{i=1}^{n} \sum_{j=1}^{n} \sigma_{i,j} x_i x_j, \tag{1}$$

$$\text{s.t.} \sum_{i=1}^{n} \mu_i x_i \geq R, \tag{2}$$

$$\sum_{i=1}^{n} x_i = 1, \tag{3}$$

$$0 \leq x_i \leq 1, i = 1, 2, \ldots, n, \tag{4}$$

where $R$ is the target return, $\mu_i$ is the expected return of the ith asset, and $\sigma_{i,j} = \rho_{i,j} \sigma_i \sigma_j$ is the covariance between the ith and jth asset. Expression (1) indicates the objective is to minimize risk, where variance is taken as the risk measure. Constraint (2) tells that minimum required return is $R$. Constraint (3) is called the budget constraint, meaning all the budget must be spent in investment. Constraint (4) means the budget share is never negative, so that short selling is not allowed. (1)–(4) altogher is called the unconstrained MV model.

#### 2.1.2. Constrained MV Model

Formally, the constrained MV model is

$$\min \gamma \sum_{i=1}^{n} \sum_{j=1}^{n} \sigma_{i,j} x_i x_j - (1 - \gamma) \sum_{i=1}^{n} \mu_i x_i \tag{5}$$

$$\text{s.t.} \sum_{i=1}^{n} x_i = 1, \tag{6}$$

$$\sum_{i=1}^{n} z_i = K, \tag{7}$$

$$\varepsilon_i z_i \leq x_i \leq \delta_i z_i, i = 1, 2, \ldots, n, \tag{8}$$

$$0 \leq x_i \leq 1, i = 1, 2, \ldots, n, \tag{9}$$

where $z_i \in \{0, 1\}$. $z_i = 1$ iff the ith asset is in the portfolio. The variable $\gamma$ in expression (5) is the risk attitude parameters. The closer $\gamma$ to 1, the more risk aversity occurs. On the other hand, the closer $\gamma$ to 0, the more risk seeking occurs. So, instead of finding the optimal values to just one value of target return, we will find a set of pareto optimal solutions. Constraint (6) is still the same budget constraint. Constraint (7) is called the cardinality constraint, i.e., the number of assets in the portfolio must be $K$. Constraint (8) is called the

quantity constraint. It is a conditional bound that limits the maximum and minimum of allocation in the individual asset if the asset is in the portfolio.

### 2.2. Biogeography-Based Optimization

BBO was first introduced by Simon (2008). Biogeography studies the distribution of species over habitats. Biogeography mathematically models the natural process of how species migrate from a habitat to another habitat, how new species emerge, and how species become extinct. Migration consists of two kinds: immigration is the event of new species entering a new habitat and emigration is the event of species leaving a habitat (but not necessarily fully disappear from the original habitat). Each habitat (mathematically expressed as $n$-dimensional vectors) has its own characteristics. Each characteristic is called suitability index variable (SIV). Basically, SIV are the decision variables in the optimization problem. Habitat suitability index (HSI) is the fitness value of a habitat. HSI measures how supportive a habitat accomodates its species. In optimization problems, HSI is the objective function. Each habitat is ranked based on the HSI, the habitat with higher HSI is ranked better than habitat with lower HSI.

Let $S_1, S_2, \ldots, S_N$ denote $N$ habitats. Habitat $S_i$ has exactly $i$ distinct species. The $d$th SIV ($d$th-coordinate) of habitat $S_i$ will be denoted by $S_i(d)$. Each habitat has its own immigration rate ($\lambda_i$) and emigration rate ($\mu_i$). Frequently, linear immigration and emigration rates are used. Guo et al. (2014) listed some of popular migration rate models, such as constant, linear, trapezoidal, quadratic, and sinusoidal models. Each migration model has their own advantages and disadvantages. In addition, they also studied the convergence properties of the BBO algorithm and some techniques on how to improve the convergence of BBO algorithm based on the migration rates. Let $E$ and $I$ denote the maximum emigration rate and immigration rate, respectively, In linear migration rate, we have

$$\lambda_i = I\left(1 - \frac{i}{n}\right) \text{ and } \mu_i = E\left(\frac{i}{n}\right), i = 0, 1, \ldots, N. \tag{10}$$

This is depicted in Figure 1. High number of species will cause a habitat saturated, resulting its species to search for better habitat and prevent more species to enter the overcrowded habitat. Hence, habitat with big number of species will have big emigration rate and small immigration rate. On the contrary, habitat will a small number of different species has big immigration rate and big emigration rate.

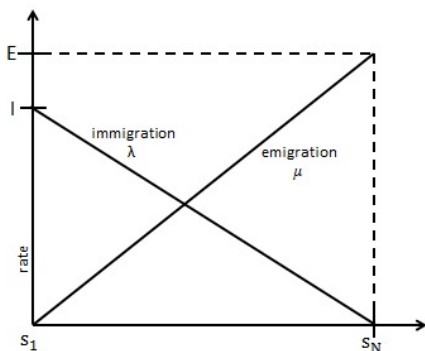

**Figure 1.** Immigration rate and Emigration rate.

In BBO, habitat with better HSI will have more species, because it is more suitable to live in. In conclusion, a better habitat will affect worse habitat to become better. The migration process in BBO is given by Algorithm 1.

---

**Algorithm 1:** Migration

---

1.  Input
    $n$, the dimension of the decision variables
    $S_k$, current solution
    $\mu$, emigration rate
    $\lambda$, immigration rate
2.  Process
    Step 1. let $i = 1$
    Step 2. If $rand(0,1) > \lambda_k$, go to step 5.
    Step 3. Choose a random habitat $S_l \sim \mu_l$.
    Step 4. Replace the $i$th coordinate of $S_k$ with the $i$th coordinate of $S_l$, $S_k(i) = S_l(i)$.
    Step 5. Set $i = i + 1$.
    Step 6. If $i > n$ stop. Otherwise, go to step 3.
3.  Output
    Migrated solution $S_k$.

---

Another variation of migration process was introduced by Ma and Simon (2011). It has a blending parameter $\alpha$ which makes migration process more random. $\alpha$ can be a constant or randomly chosen at each iteration. In this paper, we use constant $\alpha$. Algorithm 2 describes blended migration algorithm.

---

**Algorithm 2:** Blended Migration

---

1.  Input
    $n$, the dimension of the decision variables
    $S_k$, current solution
    $\mu$, emigration rate
    $\lambda$, immigration rate
    $\alpha \in [0,1]$, blending parameter.
2.  Process
    Step 1. Set $i = 1$
    Step 2. If $rand(0,1) > \lambda_k$, go to step 5.
    Step 3. Choose a random habitat $S_l \sim \mu_l$.
    Step 4. Perform a blending of immigrating and emigrating habitat using
    $S_k(i) = \alpha S_k(i) + (1 - \alpha)S_l(i)$.
    Step 5. Set $i = i + 1$.
    Step 6. If $i > n$ stop. Otherwise, go to step 3.
3.  Output
    Migrated solution $S_k$.

---

Natural events like disaster or pandemic can also happen in a habitat. It can trigger some unpredictable changes in the habitat. In BBO, this is called the mutation process. Mutation is more likely to occur in a habitat with extremely high or low number of species. On the other hand, mutation is improbable in a habitat with medium number of species. The mutation probability of a habitat is

$$m_i = m_{max}\left(1 - \frac{P_i}{P_{max}}\right). \tag{11}$$

$P_i$ denote the probability that a habitat will have species count $i$ and $P_{max} = max\{P_i | i = 1, 2, \ldots, N\}$. According to Wei et al. (2022), $P_i$ is given by

$$P_i = \begin{cases} \dfrac{1}{1+\sum_{k=1}^{N}\frac{\Pi_{l=1}^{k}\lambda_{l-1}}{\Pi_{l=1}^{k}\mu_l}}, & i = 0 \\[4mm] \dfrac{\Pi_{l=1}^{i}\lambda_{l-1}}{\Pi_{l=1}^{i}\mu_l}P_0, & i = 1, 2, \ldots, N \end{cases} \tag{12}$$

Mutation in BBO is performed based on Algorithm 3.

---

**Algorithm 3:** Mutation

---

1. Input
   $n$, the dimension of the decision variables
   $S_k$, current solution
   $m$, mutation probability
2. Process
   Step 1. Set $i = 1$
   Step 2. If $rand(0, 1) > m_k$, go to step 4.
   Step 3. Replace the $i$th coordinate of $S_k$ with a random number inside permissible bounds, $S_k(i) = rand(lowerbound, upperbound)$.
   Step 4. Let $i = i + 1$.
   Step 5. If $i > n$ stop. Otherwise, go to step 3.
3. Output
   Mutated solution $S_k$.

---

Another essential component in the BBO algorithm is the existence of elitism. In short, elitism is the preservation of optimal habitats. During the elitism process, best habitas (typically a very small subset of high ranked habitats) from previous generation will replace the worst habitats in next generation. This procedure ensures next generation best habitats are not worse than the previous generation best habitats. In their study, Ma et al. (2014) proved an interesting result about the convergence of the BBO algorithm for binary problem. The existence of migration and mutation operators without elitism do not guarantee convergence of the BBO algorithm. However, migration and mutation operators combined with elitism will almost surely produce a convergent solution. Hence, elitism is an essential part of the BBO algorithm. In general, the BBO algorithm follows Algorithm 4.

---

**Algorithm 4:** Biogeography-Based Optimization

---

1. Input
   $MaxGen$, maximum number of generations
   $N$, number of habitat
   $n$, dimension of the decision variables
   $E$, maximum emigration rate
   $I$, maximum immigration rate
   $m_{max}$, maximum mutation rate
   $keep$, number of preserved habitats
   $\alpha$, blending parameter if Blended BBO is used
2. Process
   Step 1. Initialise randomly $N$ habitats in $\mathbb{R}^n$.
   Step 2. Evaluate the HSI of each habitat and sort the habitats according HSI.
   Step 3. Calculate $\lambda, \mu$, and $m$.
   Step 4. Set $iter = 1$.
   Step 5. For $i = 1, 2, \dots, N$, perform migration according to Algorithm 1 or 2. Also, perform mutation according to Algorithm 3.
   Step 6. Sort the new habitats based on HSI.
   Step 7. Replace the last $keep$ habitats with the first $keep$ habitats from previous generation and resort the new population.
   Step 8. $iter = iter + 1$. If $iter > maxgen$, stop. Otherwise, go to step 5.
3. Output
   Return $S_{best}$ (habitat with best HSI) as the best solution.

---

### 2.3. Modified BBO

One technique to solve a constrained optimization problem is using penalty function. But, excessive number of penalty functions will have negative effects in the computational complexity. It also reduces the ability to explore and exploit feasible solutions. Most the time, large amount of candidate solutions are not feasible. A large number of iterations are needed to achieve a convergent solution. For portfolio optimization, Chang et al. (2000) developed an efficient scaling algorithm to handle cardinality constraint and quantity constraint at once. Then, the algorithm is embedded inside three optimization algorithms: GA, TS, and SA. Two interesting results from their research are that the solutions produced are both accurate and the algorithms do not require much time.

We propose using the scaling algorithm inside BBO. Furthermore, we implement some heuristic ideas from Chang et al. (2000). One of the methods they used is genetic algorithm (GA) heuristics. Since GA is similar to BBO in some ways, we modify the BBO algorithm following the idea from GA heuristics. In each iteration, rather than checking all habitats one by one, we only do migration from the good habitats. The best habitat is always chosen. Another habitat is chosen randomly only from small subset of best habitats. Furthermore, migration and mutation occurs iff the $i$th asset exists in both solutions. This greatly simplifies calculation in the BBO algorithm to focus on a subset of assets exist in good portfolio. The scaling algorithm and modified BBO is given below. This strategy is better than handling the integral constraints using penalty in the objective function, because it will require a lot of resources to find a near optimal solution. For example, Febrianti et al. (2022) used a population of 50,000 elements in solving a constrained portfolio optimization consisting of only 5 assets.

Algorithm 5 aims to modify the candidate solution to satisfy the quantity constraint. The idea is to distribute proportionally the weights of assets after ensuring the lower bound is satisfied. Then, for asset weights that are bigger than the upper bound, they are set to be the upper bound. The excesses are distributed uniformly between weights not exceeding upper bounds. Finally, the objective function is evaluated on this valid portfolio allocation. Note that Step 6 modifies the inputted habitat. This step guarantees that the same function value is produced if the processed habitat is inputted again. Algorithm 5 is crucial for the efficiency of the next algorithm.

---

**Algorithm 5:** Scaling

---

1. Input
   $S$, current solution which contains $Q$, the set of $K$ different assets, and $s_i$, the value for the $i$th asset.
   $\varepsilon$, the conditional lower bounds.
   $\delta$, the conditional upper bounds.
2. Process
   Step 1. Set $L = \sum_{i \in Q} s_i$ and $F = 1 - \sum_{i \in Q} \varepsilon_i$.
   Step 2. For $i \in Q$, $w_i = \varepsilon_i + s_i \frac{F}{L}$. $w_i = 0 \, if \, i \notin Q$.
   Step 3. $R = \{j | w_j > \delta_j\}$.
   Step 4. If $R$ is not empty: set $L = \sum_{i \in Q-R} s_i$ and $F = 1 - (\sum_{i \in Q-R} \varepsilon_i + \sum_{i \in R} \delta_i)$. For $i \in Q - R$, $w_i = \varepsilon_i + s_i \frac{F}{L}$. $w_i = \delta_i$ if $i \in R$.
   Step 5. Evaluate the objective function at $w$.
   Step 6. Set $s_i = w_i - \varepsilon_i$. for $i \in Q$.
3. Output
   Returns $S$, objective function value, and $w$.

---

In step 1, we generate feasible solutions by first generating random subset consisting of $K$ elements from $\{1, 2, \ldots, n\}$. Afterwards, random weights are generated uniformly from $[0, 1]$. Afterwards, weights are normalized in order that the sum of weights equals to 1 (budget constraint). The process in Algorithm 6 basically follows the same process as Algorithm 4. The main difference is the Step 5. We choose *BestHabitats* only from small

range of indices, for example *BestHabitats* can be the set of index $[2, 3, \ldots, 10]$. Thus, only the good habitats are migrated with the best habitat (habitat with the lowest function value for minimization). This idea greatly simplifies the complexity in each iteration compared than Algorithm 4, where in Algorithm 4, all habitats are considered for migration. The selection is done via roulette wheel selection. Each habitat $S_i (l \in BestHabitats)$ are assigned probability proportional to their $\lambda_i$. Subsequently, a random number is generated to determine which habitat is picked for migration. The elitism in Algoritm 6 is also more efficient than Algorithm 4, since only one habitat is replaced in each iteration.

---

**Algorithm 6:** Modified BBO for Large Scale Portfolio Optimization

1. Input
   *MaxGen*, maximum number of generations
   $N$, number of habitats
   $n$, dimension of the decision variables
   $E$, maximum emigration rate
   $I$, maximum immigration rate
   $m_{max}$, maximum mutation rate
   *BestHabitat*, limit of the best habitats
   $\alpha$, blending parameter
   $K$, number of assets in the portfolio
2. Process
   Step 1. Initialise randomly $N$ feasible habitats in $[0, 1]^n$.
   Step 2. Evaluate and update each habitat using Algorithm 4. Sort the habitats according HSI.
   Step 3. Calculate $\lambda, \mu$, and $m$.
   Step 4. Set *iter* = 1.
   Step 5. Choose a random habitat $S_j \sim \lambda_j$ from *BestHabitats*.
   Step 6. Create a new habitat $S*$ according to blended migration of $S_{best}$ and $S_j$.
   Step 7. Perform mutation on $S*$ according to Algorithm 2.
   Step 8. If any coordinate of $S*$ is negative, replace it with 0.
   Step 9. Set the assets of $S*$ to be the location of positive coordinates. If the number of assets in $S*$ is less than $K$, then randomly add assets from $S_1$ or $S_j$. If the number of assets in $S*$ is bigger than $K$, randomly choose some assets, remove it.
   Step 10. Evaluate and update $S*$ based on Algorithm 4.
   Step 11. Replace $S_{worst}$ (habitat with worst HSI) with $S*$ and sort the habitats based on HSI.
   Step 12. Set *iter* = *iter* + 1. If *iter* > *MaxGen*, stop. Otherwise, go to step 5.
3. Output
   Return $S_{best}$ (habitat with lowest function value for minimization problem) as the best solution.

---

## 3. Results and Conclusions

In this section, we first apply the above ideas to measure its effectiveness. We consider using percentage deviation error as the measure of the model effectiveness as used in Chang et al. (2000) and Kabbani (2022). Let $(v_i, r_i)$ denote the pair of variance and return at a point in constrained efficient frontier (CEF) found using proposed method. Let also $(v_j^*, r_j^*)$ denote the point in unconstrained efficient frontier calculated by Chang et al. (2000). For each $i$, we can find $v_l^* = max\{v_j^* | v_j^* \leq v_i\}$ and $v_u^* = min\{v_j^* | v_j^* \geq v_i\}$. Then, $r = r_l^* + \frac{v_i - v_l^*}{v_u^* - v_l^*}(r_u^* - r_l^*)$ is the approximated return from UEF at $r_i$. The vertical deviation error is calculated using $\left| \frac{r_i - r}{r} \times 100 \right|$. The horizontal deviation error is calculated in a

similar fashion. The percentage deviation error is taken as the minimum of the horizontal and vertical deviation error.

### 3.1. Results

In this section we solve problem (5)–(9) for 50 values of $\gamma$. We assume $K = 10$ (10 assets are chosen for each case), $\varepsilon_i = 0.01$ (the minimum budget allocation for each chosen asset is 0.01), and $\delta_i = 1$ (maximum budget allocation is 1 for each asset). We use the same dataset as Chang et al. (2000) and Kabbani (2022). In total, there are 5 test instances studied. The data used are the Hang Seng (Hong Kong), DAX 100 (Germany), FTSE 100 (UK), S&P 100 (USA) and Nikkei 225 (Japan) index weekly returns from March 1992 to September 1997. In total, there are 5 test instances studied. For each instance, we generated 50 points, one point for each value of $\gamma$. $\gamma$ will vary from 0 to 1 uniformly with a step length of $\frac{50}{49}$. The parameters used are summarized in Table 1. The comparisons between the proposed method with previous studies are given in Table 2.

**Table 1.** Parameters in Modified BBO.

| Parameter | Value |
|:---:|:---:|
| $n$ | 31, 85, 89 98, and 225 |
| *MaxGen* | 1500n |
| $N$ | 100 |
| $E$ | 1 |
| $I$ | 1 |
| $m_{max}$ | 0.05 |
| *BestHabitats* | $\{2, 3, \ldots, 10\}$ |

**Table 2.** Comparison of performance.

| Index | Number of Assets | Error and Time | GA | TS | SA | TS&TR | BBO |
|:---:|:---:|:---:|:---:|:---:|:---:|:---:|:---:|
| Hang Seng | 31 | Median | **1.2181** | **1.2181** | **1.2181** | 1.8120 | 1.2503 |
| | | Mean | **1.0974** | 1.1217 | 1.0957 | 2.2656 | 1.1689 |
| | | Time (s) | 172 | 74 | 79 | 1154 | 282 |
| DAX | 85 | Median | **2.5466** | 2.6380 | 2.5661 | 4.2100 | 2.8845 |
| | | Mean | **2.5424** | 3.3049 | 2.9297 | 4.0350 | 2.7018 |
| | | Time (s) | 544 | 199 | 210 | 2873 | 1830 |
| FTSE | 89 | Median | **1.0841** | **1.0841** | **1.0841** | 1.2406 | 1.1232 |
| | | Mean | 1.1076 | 1.6080 | 1.4623 | 1.2959 | **1.1056** |
| | | Time (s) | 573 | 246 | 215 | 2919 | 1941 |
| S&P | 98 | Median | 1.2244 | 1.2882 | **1.1823** | 2.3630 | 1.3671 |
| | | Mean | 1.9328 | 3.3092 | 3.0696 | 2.5068 | **1.8782** |
| | | Time (s) | 638 | 225 | 242 | 3107 | 2310 |
| Nikkei | 225 | Median | 0.6133 | 0.6093 | **0.6066** | 1.3464 | 2.1840 |
| | | Mean | 0.7961 | 0.8975 | **0.6732** | 1.2122 | 2.6556 |
| | | Time (s) | 1964 | 545 | 553 | 5866 | 6382 |

All numerical experiments are done in MATLAB Online. Best numerical experiments are shown in bold.

### 3.2. Conclusions

This paper discusses the extensions of the classical MV portfolio model to fit proper real-world situations. The extensions include adding cardinality and quantity constraints which are practical for most investors. The lack of decent approaches to solve the problem, especially the large scale ones, stimulates many alternative approaches. We propose the usage of modified BBO to solve the problem. It uses some ideas from Chang et al. (2000) that handle both constraints effectively. The algorithm makes the candidate solutions

always satisfy both constraint. This causes the algorithm to yield a convergent result more quickly than letting them evolve wildly.

Tables 3–7 list the points generated by our proposed approach and their average percentage deviation. Figures 2–6 shows the unconstrained efficient frontier (UEF) and constrained efficient frontier (CEF) obtained by proposed method. We see that CEF only deviates by a small amount from UEF. From Table 2, we see that modified BBO works pretty well in large scale portfolio optimization. Although GA works best in most instances, modified BBO can still give good near-optimal solutions. Especially, in the third and fourth instances, modified BBO produce lower mean in percentage deviation error than the other methods. The performance of our proposed apporach for the last instance is the worst compared to other methods. To overcome this, more iterations can be performed to the method at the cost of computation time.

**Table 3.** First and last three optimal portfolios of Hang Seng.

| $\gamma = 0$ | | $\gamma = 0.0204$ | | $\gamma = 0.0408$ | |
|---|---|---|---|---|---|
| Chosen Asset | Weights | Chosen Asset | Weights | Chosen Asset | Weights |
| 2 | 0.0100 | 2 | 0.0100 | 4 | 0.0100 |
| 4 | 0.0100 | 5 | 0.9094 | 5 | 0.9090 |
| 5 | 0.9097 | 8 | 0.0101 | 9 | 0.0108 |
| 9 | 0.0101 | 9 | 0.0101 | 12 | 0.0100 |
| 12 | 0.0100 | 12 | 0.0100 | 15 | 0.0100 |
| 13 | 0.0100 | 15 | 0.0100 | 19 | 0.0100 |
| 20 | 0.0100 | 19 | 0.0100 | 20 | 0.0100 |
| 24 | 0.0100 | 20 | 0.0100 | 23 | 0.0101 |
| 27 | 0.0100 | 26 | 0.0100 | 26 | 0.0100 |
| 29 | 0.0102 | 29 | 0.0104 | 29 | 0.0100 |
| Return | 0.0103 | Return | 0.0103 | Return | 0.0103 |
| Risk | 0.0042 | Risk | 0.0041 | Risk | 0.0041 |

| $\gamma = 0.9592$ | | $\gamma = 0.9796$ | | $\gamma = 1$ | |
|---|---|---|---|---|---|
| Chosen Asset | Weights | Chosen Asset | Weights | Chosen Asset | Weights |
| 5 | 0.0408 | 5 | 0.0210 | 2 | 0.0146 |
| 9 | 0.0288 | 13 | 0.0434 | 13 | 0.0461 |
| 13 | 0.0323 | 15 | 0.1035 | 15 | 0.0761 |
| 15 | 0.1176 | 16 | 0.0697 | 16 | 0.1068 |
| 16 | 0.0257 | 17 | 0.0183 | 17 | 0.0476 |
| 26 | 0.1693 | 26 | 0.1583 | 26 | 0.1437 |
| 28 | 0.2825 | 28 | 0.2965 | 28 | 0.3044 |
| 29 | 0.1688 | 29 | 0.1195 | 29 | 0.0636 |
| 30 | 0.0870 | 30 | 0.1169 | 30 | 0.1335 |
| 31 | 0.0472 | 31 | 0.0530 | 31 | 0.0635 |
| Return | 0.0103 | Return | 0.0034 | Return | 0.0028 |
| Risk | 0.0042 | Risk | $6.4848 \times 10^{-4}$ | Risk | $6.4228 \times 10^{-4}$ |

**Table 4.** First and last three optimal portfolios of DAX.

| $\gamma = 0$ | | $\gamma = 0.0204$ | | $\gamma = 0.0408$ | |
|---|---|---|---|---|---|
| Chosen Asset | Weights | Chosen Asset | Weights | Chosen Asset | Weights |
| 2 | 0.0102 | 2 | 0.0108 | 2 | 0.0100 |
| 11 | 0.0101 | 11 | 0.0100 | 13 | 0.0151 |
| 13 | 0.0110 | 13 | 0.0120 | 29 | 0.0118 |
| 15 | 0.0100 | 29 | 0.0127 | 37 | 0.0100 |
| 29 | 0.0141 | 38 | 0.9043 | 38 | 0.9031 |
| 30 | 0.0100 | 41 | 0.0100 | 41 | 0.0100 |
| 37 | 0.0112 | 47 | 0.0100 | 43 | 0.0100 |
| 38 | 0.9032 | 57 | 0.0100 | 46 | 0.0100 |
| 49 | 0.0101 | 74 | 0.0101 | 47 | 0.0100 |
| 74 | 0.0102 | 77 | 0.0100 | 69 | 0.0100 |
| Return | 0.0093 | Return | 0.0093 | Return | 0.0093 |
| Risk | 0.0024 | Risk | 0.0024 | Risk | 0.0024 |

**Table 4.** *Cont.*

| γ = 0.9592 | | γ = 0.9796 | | γ = 1 | |
|---|---|---|---|---|---|
| Chosen Asset | Weights | Chosen Asset | Weights | Chosen Asset | Weights |
| 2 | 0.0952 | 2 | 0.0915 | 2 | 0.0711 |
| 4 | 0.1123 | 4 | 0.1880 | 4 | 0.2193 |
| 13 | 0.1490 | 13 | 0.0966 | 12 | 0.0450 |
| 15 | 0.0730 | 19 | 0.0538 | 13 | 0.0541 |
| 29 | 0.1015 | 29 | 0.0565 | 19 | 0.0993 |
| 38 | 0.0573 | 49 | 0.1292 | 49 | 0.1103 |
| 49 | 0.1221 | 51 | 0.0594 | 51 | 0.0818 |
| 57 | 0.0731 | 59 | 0.0666 | 59 | 0.0664 |
| 68 | 0.1462 | 68 | 0.1759 | 68 | 0.1773 |
| 71 | 0.0703 | 71 | 0.0825 | 71 | 0.0747 |
| Return | 0.0047 | Return | 0.0033 | Return | 0.0025 |
| Risk | $1.9717 \times 10^{-4}$ | Risk | $1.5951 \times 10^{-4}$ | Risk | $1.4872 \times 10^{-4}$ |

**Table 5.** First and last three optimal portfolios of FTSE.

| γ = 0 | | γ = 0.0204 | | γ = 0.0408 | |
|---|---|---|---|---|---|
| Chosen Asset | Weights | Chosen Asset | Weights | Chosen Asset | Weights |
| 1 | 0.0101 | 10 | 0.0126 | 5 | 0.0100 |
| 2 | 0.0102 | 18 | 0.9056 | 9 | 0.0103 |
| 6 | 0.0100 | 19 | 0.0100 | 10 | 0.0113 |
| 9 | 0.0100 | 29 | 0.0114 | 18 | 0.9042 |
| 10 | 0.0100 | 44 | 0.0101 | 19 | 0.0100 |
| 18 | 0.9038 | 55 | 0.0100 | 26 | 0.0100 |
| 29 | 0.0133 | 66 | 0.0100 | 29 | 0.0141 |
| 37 | 0.0124 | 71 | 0.0104 | 44 | 0.0100 |
| 55 | 0.0100 | 72 | 0.0100 | 55 | 0.0100 |
| 82 | 0.0100 | 76 | 0.0100 | 68 | 0.0100 |
| Return | 0.0079 | Return | 0.0079 | Return | 0.0079 |
| Risk | 0.0013 | Risk | 0.0013 | Risk | 0.0013 |
| γ = 0.9592 | | γ = 0.9796 | | γ = 1 | |
| Chosen Asset | Weights | Chosen Asset | Weights | Chosen Asset | Weights |
| 2 | 0.1848 | 2 | 0.1534 | 2 | 0.1264 |
| 25 | 0.0688 | 25 | 0.0987 | 20 | 0.0812 |
| 30 | 0.1028 | 30 | 0.0828 | 25 | 0.0949 |
| 41 | 0.0555 | 41 | 0.0922 | 30 | 0.0707 |
| 46 | 0.0911 | 46 | 0.1484 | 41 | 0.1182 |
| 53 | 0.0943 | 53 | 0.0728 | 45 | 0.0524 |
| 62 | 0.2044 | 62 | 0.1709 | 46 | 0.1556 |
| 66 | 0.0634 | 71 | 0.0286 | 62 | 0.1387 |
| 72 | 0.0524 | 75 | 0.0795 | 75 | 0.0810 |
| 82 | 0.0826 | 83 | 0.0727 | 83 | 0.0808 |
| Return | 0.0038 | Return | 0.0033 | Return | 0.0025 |
| Risk | $2.3204 \times 10^{-4}$ | Risk | $2.1689 \times 10^{-4}$ | Risk | $2.0841 \times 10^{-4}$ |

Possible improvements can be made to the proposed method, such as using ideas of set-based metaheuristic algorithms to take care of cardinality and quantity constraints separately. Such approach has been studied by some researches, for instance see Erwin and Engelbrecht (2020). The idea is to choose a certain subset of assets first, then optimize the portfolio allocation for each subsets. Looking at various migration and mutation models are also interesting direction to do.

**Table 6.** First and last three optimal portfolios of S&P.

| $\gamma = 0$ | | $\gamma = 0.0204$ | | $\gamma = 0.0408$ | |
|---|---|---|---|---|---|
| Chosen Asset | Weights | Chosen Asset | Weights | Chosen Asset | Weights |
| 20 | 0.0100 | 12 | 0.0100 | 2 | 0.0100 |
| 31 | 0.0100 | 14 | 0.0103 | 14 | 0.0104 |
| 34 | 0.0119 | 20 | 0.0101 | 23 | 0.0100 |
| 36 | 0.0100 | 34 | 0.0256 | 34 | 0.0266 |
| 42 | 0.0104 | 42 | 0.0108 | 42 | 0.0103 |
| 43 | 0.0100 | 55 | 0.0100 | 55 | 0.0100 |
| 82 | 0.9075 | 56 | 0.0100 | 67 | 0.0100 |
| 85 | 0.0100 | 82 | 0.8897 | 82 | 0.8878 |
| 89 | 0.0101 | 86 | 0.0100 | 89 | 0.0149 |
| 96 | 0.0100 | 89 | 0.0135 | 93 | 0.0100 |
| Return | 0.0089 | Return | 0.0089 | Return | 0.0089 |
| Risk | 0.0025 | Risk | 0.0025 | Risk | 0.0025 |
| $\gamma = 0.9592$ | | $\gamma = 0.9796$ | | $\gamma = 1$ | |
| Chosen Asset | Weights | Chosen Asset | Weights | Chosen Asset | Weights |
| 11 | 0.0844 | 11 | 0.0894 | 10 | 0.0604 |
| 19 | 0.0789 | 34 | 0.0392 | 19 | 0.0895 |
| 34 | 0.0520 | 36 | 0.0635 | 28 | 0.0648 |
| 36 | 0.0923 | 37 | 0.1459 | 33 | 0.0568 |
| 45 | 0.1751 | 62 | 0.2074 | 37 | 0.1504 |
| 52 | 0.0578 | 64 | 0.0703 | 51 | 0.0797 |
| 62 | 0.1713 | 65 | 0.1131 | 62 | 0.2530 |
| 64 | 0.0767 | 73 | 0.0603 | 64 | 0.0763 |
| 86 | 0.1047 | 86 | 0.1003 | 65 | 0.1001 |
| 96 | 0.1067 | 96 | 0.1106 | 73 | 0.0691 |
| Return | 0.0035 | Return | 0.0028 | Return | 0.0018 |
| Risk | $2.3204 \times 10^{-4}$ | Risk | $1.4646 \times 10^{-4}$ | Risk | $1.3461 \times 10^{-4}$ |

**Table 7.** First and last three optimal portfolios of Nikkei.

| $\gamma = 0$ | | $\gamma = 0.0204$ | | $\gamma = 0.0408$ | |
|---|---|---|---|---|---|
| Chosen Asset | Weights | Chosen Asset | Weights | Chosen Asset | Weights |
| 2 | 0.0100 | 9 | 0.0247 | 2 | 0.0104 |
| 40 | 0.0103 | 40 | 0.0103 | 9 | 0.0119 |
| 62 | 0.0145 | 62 | 0.0100 | 115 | 0.0134 |
| 68 | 0.0100 | 79 | 0.0100 | 137 | 0.0100 |
| 97 | 0.0100 | 104 | 0.0100 | 165 | 0.0105 |
| 115 | 0.0101 | 114 | 0.0100 | 186 | 0.0100 |
| 154 | 0.0100 | 115 | 0.0138 | 212 | 0.0100 |
| 186 | 0.0100 | 165 | 0.0379 | 214 | 0.8937 |
| 201 | 0.0100 | 201 | 0.0101 | 215 | 0.0201 |
| 214 | 0.9051 | 214 | 0.8631 | 224 | 0.0100 |
| Return | 0.0038 | Return | 0.0038 | Return | 0.0038 |
| Risk | 0.0015 | Risk | 0.0014 | Risk | 0.0015 |
| $\gamma = 0.9592$ | | $\gamma = 0.9796$ | | $\gamma = 1$ | |
| Chosen Asset | Weights | Chosen Asset | Weights | Chosen Asset | Weights |
| 8 | 0.0110 | 60 | 0.1766 | 11 | 0.0802 |
| 40 | 0.0100 | 97 | 0.0100 | 60 | 0.2004 |
| 60 | 0.2729 | 98 | 0.1337 | 62 | 0.1656 |
| 62 | 0.2386 | 114 | 0.0500 | 97 | 0.0141 |
| 97 | 0.1364 | 129 | 0.2653 | 98 | 0.1519 |
| 104 | 0.0102 | 162 | 0.0435 | 105 | 0.0617 |
| 129 | 0.0743 | 165 | 0.1810 | 129 | 0.1305 |
| 158 | 0.0100 | 196 | 0.1143 | 144 | 0.0100 |
| 171 | 0.1309 | 215 | 0.0103 | 199 | 0.0100 |
| 225 | 0.1058 | 225 | 0.0152 | 225 | 0.1756 |
| Return | $8.4069 \times 10^{-4}$ | Return | $4.6038 \times 10^{-4}$ | Return | $2.5560 \times 10^{-5}$ |
| Risk | $3.4432 \times 10^{-4}$ | Risk | $3.4888 \times 10^{-4}$ | Risk | $3.1395 \times 10^{-4}$ |

Another possible future research is to consider some more practical constraints such as roundlot constraint, transaction cost, preselection constraint, and tracking error constraint. The idea of putting in more constraints is so that investor can fully realizes their investment plans. Overall, the performance of the proposed method is satisfying in solving large scale constrained portfolio optimization.

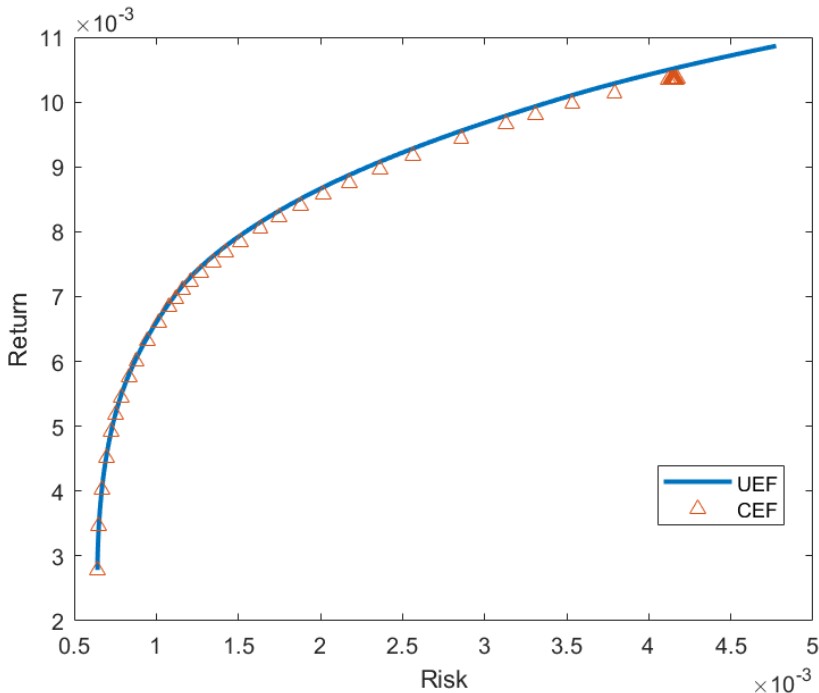

**Figure 2.** Constrained Efficient Frontier for Hang Seng.

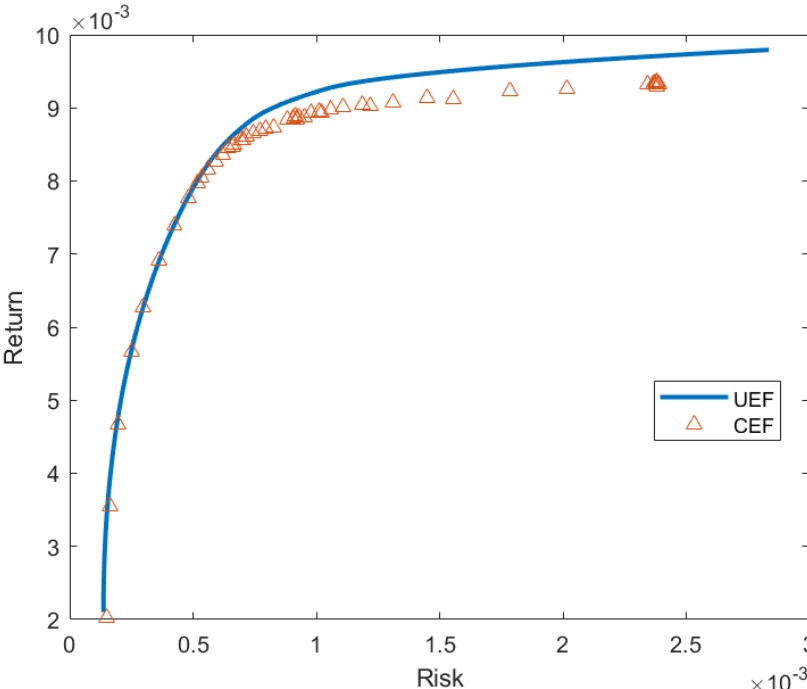

**Figure 3.** Constrained Efficient Frontier for DAX.

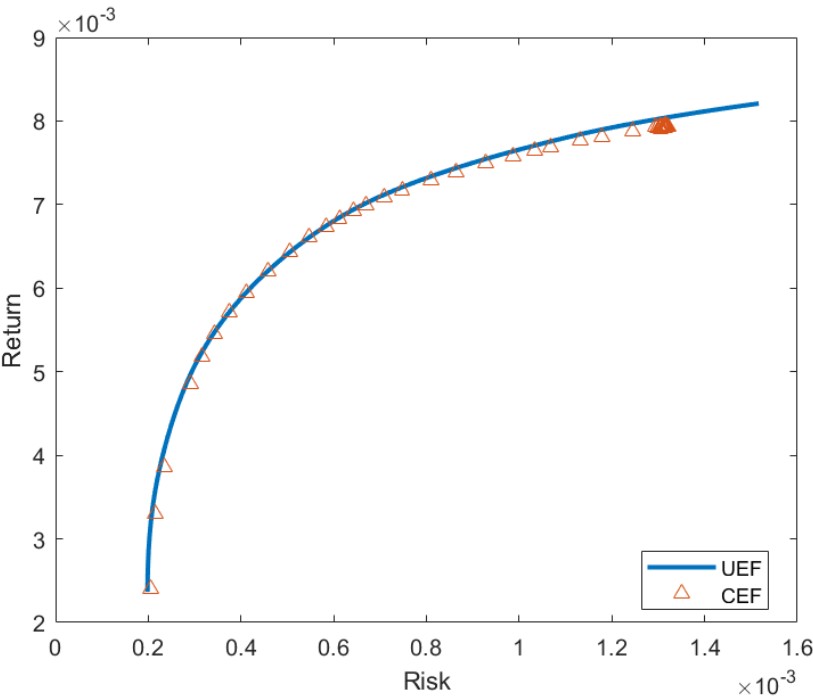

**Figure 4.** Constrained Efficient Frontier for FTSE.

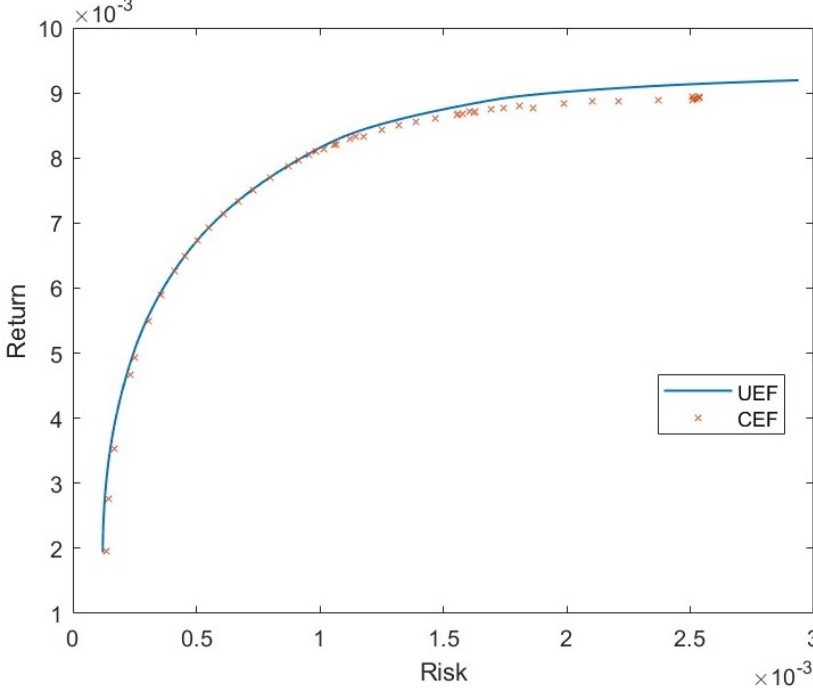

**Figure 5.** Constrained Efficient Frontier for S&P.

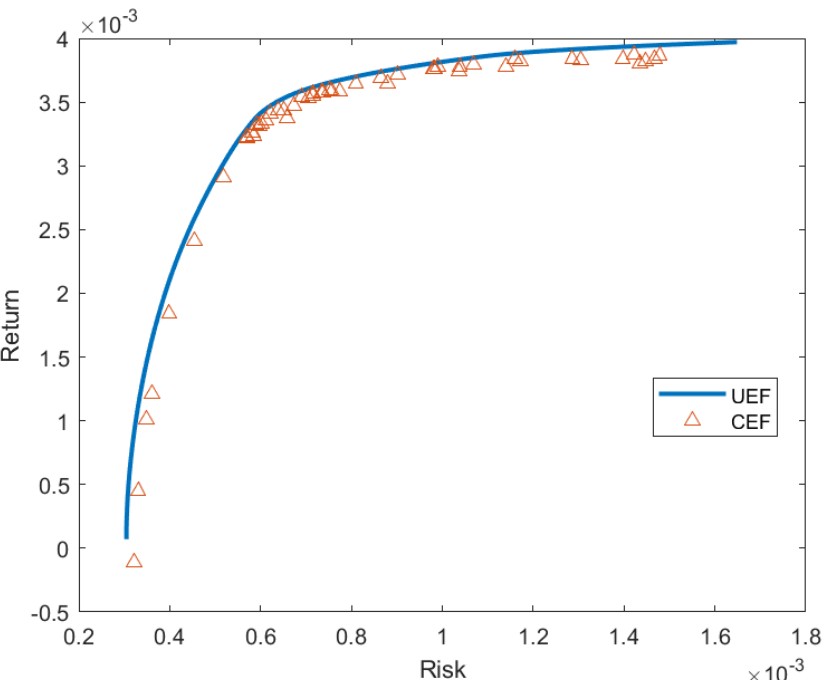

**Figure 6.** Constrained Efficient Frontier for Nikkei.

**Author Contributions:** Supervision, K.A.S.; Writing—original draft, W.W. All authors have read and agreed to the published version of the manuscript.

**Funding:** This reseach is not externally funded.

**Informed Consent Statement:** Not applicable.

**Data Availability Statement:** The data used in this paper are available at http://people.brunel.ac.uk/~mastjjb/jeb/orlib/files/, accessed on 24 April 2023.

**Conflicts of Interest:** The authors declare no conflict of interest.

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
