# Peer review of "Large-Scale Portfolio Optimization Using Biogeography-Based Optimization"

_ijfs, doi:10.3390/ijfs11040125_

Round 1
Reviewer 1 Report
Comments and Suggestions for Authors
The problem here was the Markowitz problem with a no short sale constraint. As the authors point out, adding large numbers of assets to the problem makes it hard to solve. There are many solutions to this many asset problem in the finance literature
1) The capital asset pricing model, which also includes the insight that really there is one market portfolio (two fund separation) -- t
2) tracking error to the market portfolio -- empirically when there are more than 30 or so assets the properties of the portfolio is hard to distinguish from a market index -- a large literature around tracking error has built up here.
3) The ability to rebalance portfolios at intermediate times -- rather than a single time "set and forget" portfolio as described here.
None of tese aspects were discussed in the literature review. Instead the literature review is a bit idiosyncratic. I suggest citing a review paper or text.
However, I am confident that the novelty of the paper -- of adding biogeographical optimization -- is there. That optimization method was described for the original problem in population biology, but I didn't understand exactly how they authors adapted the problem to the portfolio optimization setting. That needs to be better described before the paper can be published.
I was not surprised that the results showed little difference in efficient frontiers with our without the additional constraints, but it was interesting that the less liquid market (Hang Seng) had constraints that mattered more than the more liquid markets.
Note that the returns here are in thousanths. I expect that daily returns and vols are being used. All quantities are typically annualized in this kind of paper. The output of the figures is not incorrect but it would be a bit like quoting highway traffic speeds at m/sec and expecting the drivers to translated their speedometer readings of km/h into m/sec. That is in some sense trivial but we wouldn't expect drivers to do it.
The tables of points should be left out and made available to readers on request. Or put in an online appendix if available by this journal. Otherwise they add little.
I was also not clear on how the variance - covariance matrices for each market were assembled. Estimation error for these matrices (or non stationarity) typically overwhelm the details of the optimization method used. (Garlappi, Uppal, De Miguel or Grauer and Best).
In any case the authors should describe where the variance covariance matrices came from.
Comments on the Quality of English Language
The English is more or less fine. There are a few grammatical errrors and even a spelling mistake or two -- a good proofread by is worth buying -- but the English never stopped me from understanding what the authors were doing (sometimes I didn't understand, but that's because they didn't explain it at all, not because the English was faulty).
Author Response
- Added CAPM reference in introduction and why it is not suitable for problem discussed. Tracking error constraint is added as future direction in conclusion. Rebalancing is not discussed in the paper since we only consider one step ahead budget allocation.
- The algorithm is adapted to solve portfolio optimization because the reasons developed in introduction. The algorithm showed promising results. In the modification itself, the algorithm makes sure the results are feasible portfolios, due to constraints added. Explanations on algorithms used are added (for algorithm 4 and 5).
- The returns and variances are not in percentages. The only one in percentage is the error deviation.
- The data used are secondary data from the resource stated in "Data availability". The raw data are not available to us, so we can't say much about other things than the ones listed. The data contains mean returns, variances, and correlations of each stock considered. From there, the covariance matrix are assembled using cov(X,Y) = rho(X,Y) * sd(X) * sd(Y).
Reviewer 2 Report
Comments and Suggestions for Authors
This paper is dedicated to the advancements of the genetic algorithms (GA) for the mean-variance portfolio optimization (MVO). The authors introduce an improvement to the BBO subset of GA, which is one of the most efficient way to address the constrained MVO problems. GA provides several important advantages over classic MVO, such as gradient quadratic programming algorithms, for example, they are not prone to local optimality and also do not depend of the derivative of the objective function. That's why advancements in GA represent valuable contribution to practical portfolio management.
Unfortunately, most investment professionals are not familiar with GA and most academics who develop GA fail to provide sound cases for their application to portfolio management, so their usage is limited. Overcoming this drawback is, in my opinion, an important task, and this paper provides an essential step, so I find it relevant, interesting, and useful. I recommend it for publishing after addressing some minor issues listed below:
1. Lines 131 and 205 contain "??" or "?" instead of proper references.
2. The authors did not clarify how they obtain a feasible initial solution. The current version of their MATLAB code suggests assigning random weights, but that only works if no additional constraints besides (6) exists.
3. The authors fail to explain Step 2.5 in Algorithm 5 (line 289) - how they choose a random habitat from BestHabitats. From the MATLAB code I see they choose it from habitats 2 to 10 using RouletteWheelSelection(), but the function RouletteWheelSelection() is never disclosed or explained in the paper.
4. In section 3 results are evaluated only on their deviation from the unconstrained efficient frontier. I do not think it makes much sense. In my opinion, if the resulting portfolio are close enough in MV space it does not matter much, but the authors fail to disclose the time required to run the algorithms. In their MATLAB code I see their account for the running time, and it's worth disclosing this information to show computational efficiency.
5. When the authors run different GA algorithms, including the proposed one, on samples of different assets that are constituents of major stock indices (S&P 500, DAX, FTSE 100, and so on) they do not disclose important information on the data samples used and the resulting portfolios. That information would be crucial for the investment professionals to understand the outcomes of the algorithms. What is good to have for each data sample: time period (from, to) of price or returns data, price or return data provider, what prices (raw or adjusted for dividends / stock splits) were used. For the results, I would be good to see 5-6 optimal portfolios disclosed (assets, risk, return) for different values of Gamma (e.g. 0, 0.2, 0.4, 0.6, 0.8, 1).
Having only limited information that can be derived for the supplements I was only able to check the S&P optimal solution against the gradient MVO problem solver, and it seems to generate a comparable portfolio in MV space. However, due to lack of information the authors provided about their data samples I wasn't able to perform full-scale checks of other portfolios.
Areas of additional improvement of this paper may include:
1. The authors only use one constraint (6) on assets weight, however in practical application a large number of linear constraints are used to account for sector or geography diversification. How could such constrains be added to BBO GA?
2. The authors only consider fixed number of assets (10 in this case), but in real application investors mostly interested in keeping this number reasonable, but never a fixed value. I.e. for small investor from 5 to 15 assets is OK, for an institution somewhat from 20-30 to 50-60 assets in a portfolio. Would the authors consider having flexible bounds (e.g. lower 5, upper 15) for the numR variable instead of fixed value 10 they used?
Author Response
- Fixed ? and ?? by adding suitable citations.
- Added explanations how to get initial feasible solutions (after algorithm 5).
- bestHabitats are defined as set of indices with good HSI (after ranked). Added clarifications.
- The original and new reference paper (2022) also used the same model accuracy measure. So, to make fair comparisons, we used the same model accuracy measure also.
- Added optimal portfolios as requested.
- We would like to thank you for your kind suggestions such as dynamic portfolio sizes and considering diversification for the portfolio selection. These are wonderful ideas and we will consider it next time for our researches.